# Is Systemic Immunosuppression a Risk Factor for Oral Cancer? A Systematic Review and Meta-Analysis

**DOI:** 10.3390/cancers15123077

**Published:** 2023-06-06

**Authors:** Romeo Patini, Massimo Cordaro, Denise Marchesini, Francesco Scilla, Gioele Gioco, Cosimo Rupe, Maria Antonietta D’Agostino, Carlo Lajolo

**Affiliations:** 1Department of Head, Neck and Sense Organs, School of Dentistry, Catholic University of Sacred Heart, Fondazione Policlinico Universitario “A. Gemelli”—IRCCS Rome, 00135 Rome, Italy; romeo.patini@unicatt.it (R.P.); massimo.cordaro@unicatt.it (M.C.); denise.marchesini01@icatt.it (D.M.); francesco.scilla01@icatt.it (F.S.); cosimorupe@gmail.com (C.R.); carlo.lajolo@unicatt.it (C.L.); 2Department of Geriatric and Orthopedic Sciences, Catholic University of Sacred Heart, Fondazione Policlinico Universitario “A. Gemelli”—IRCCS Rome, 00135 Rome, Italy; mariaantonietta.dagostino@unicatt.it

**Keywords:** immunosuppression, oral cancer, systematic review, meta-analysis

## Abstract

**Simple Summary:**

Immunosuppression is a medical condition in which a person’s immune system is unable to function properly, or it does not function at all. It is a well-known fact that an ill-functioning immune system can favor the generation and development of potentially malignant lesions, autoimmune and allergic diseases, and even neoplasms. At present, the amount of risk for the development of oral cancer in immunosuppressed patients has not been quantitatively reported. Such a topic has been investigated, revealing that immunosuppression increases the risk of developing cancer from 0.2% to 1% (95% CI: 0.2% to 1.4%), giving further importance to the accurate follow-up of this category of patients.

**Abstract:**

Even if the relationship between immunosuppression and increased incidence of systemic cancers is well known, there is less awareness about the risk of developing oral cancer in immunosuppressed patients. The aim of this review was to evaluate the association between immunosuppression and the development of oral cancer. Two authors independently and, in duplicate, conducted a systematic literature review of international journals and electronic databases (MEDLINE via OVID, Scopus, and Web of Science) from their inception to 28 April 2023. The assessment of risk of bias and overall quality of evidence was performed using the Newcastle–Ottawa Scale and GRADE system. A total of 2843 articles was identified, of which 44 met the inclusion criteria and were included in either the qualitative or quantitative analysis. The methodological quality of the included studies was generally high or moderate. The quantitative analysis of the studies revealed that immunosuppression should be considered a risk factor for the development of oral cancer, with a percentage of increased risk ranging from 0.2% to 1% (95% CI: 0.2% to 1.4%). In conclusion, the results suggest that a constant and accurate follow-up should be reserved for all immunosuppressed patients as a crucial strategy to intercept lesions that have an increased potential to evolve into oral cancer.

## 1. Introduction

According to official data from the World Health Organization (WHO, Geneva, Switzerland), 377,713 new cases of oral and lip cancer were diagnosed in 2020, making it the 16th most common cancer in the world. It still has a severe prognosis today, as approximately 50% of oral and lip cancer patients will die in the 5 years following diagnosis, while the remaining 50% have aesthetic and functional relics that make their quality of life rather low. Historically, the main risk factors for this neoplasm are being male, having a diet low in vitamins, having MPDs, past/present viral infections, radiation exposure, having genetic predispositions and immunodeficiencies, and engaging in luxuriant habits such as smoking and alcohol and betel consumption [1].

Oral cancer treatment is challenging and requires a multidisciplinary approach with a team of specialists, which includes head and neck surgeons, radiation oncologists, medical oncologists, and oral oncologists [2].

Although surgery is the most common initial definitive treatment for the majority of oral cancers, adjunctive radiotherapy (RT), with or without chemotherapy (CT) may be performed [3].

The immune system performs numerous functions, among which its primary functions are defense against infections, self-control and immunosurveillance at the onset and during the proliferation of solid and liquid cancers, identifying and suppressing genetically modified cells that have already passed the normal checkpoints, and the intracellular control of proliferation. The possible role of the immune system in the development of cancers has been defined in the theory of “immune surveillance”, which configures the active role of the immune system in preventing the onset of cancers [4].

Immune surveillance against cancer is the process in which the immune system identifies cancerous and/or precancerous cells and eliminates them.

According to the most recent findings, the immune system can play a role in preventing tumors, throughout different mechanisms. First, the virus-induced tumors can be prevented when a functioning immune system can eliminate or suppress viral infections. Second, this action against pathogens may cause a prompt resolution of inflammation, preventing the establishment of an inflammatory environment, which is a risk factor for carcinogenesis [5]. Third, the immune system can identify and eliminate tumor cells on the basis of their expression of tumor-specific antigens. Therefore, the theory of immunosurveillance is essentially based on two generally accepted claims: (I) most cancers are antigenic (an obvious requirement for immunological recognition) and (II) such antigenic differences can, “under appropriate conditions”, elicit an immune response [4].

Despite immune surveillance, cancers develop even in the presence of a functioning immune system, and therefore, currently, we speak of “cancer immunoediting”, a term which is used to describe the evolution of tumors, wherein tumor cells become less effectively recognized and killed by the immune system [6,7].

A first consideration concerns the definition that is used for patients with disorders of the immune system. The terms immunosuppression and immunodeficiency are often used interchangeably. This confusion is related to the subtle nuance that separates them. It could be specified that immunosuppression identifies a medical condition of a general malfunction of the immune system. Immunodeficiency, on the other hand, classifies the severity of this physical deficit according to two categories: primary and secondary.

Immunosuppression is a pathological condition characterized by the inhibition of one or more components of the immune system, whether natural or acquired, resulting in the impossibility of a person’s immune system to function properly. However, currently, there is no description illustrating the relationship between immunoediting and immunosuppression. The incorrect functioning of the immune system can favor the development of autoimmune and allergic diseases or neoplasms. Immunodeficiencies are divided into primary (if they are derived from congenital defects) and secondary (if they are derived from infections or pharmacological treatments) classifications. This condition involves the onset of infections that develop and recur very often, manifesting themselves in a more serious and longer-lasting form.

Among the many alterations of the immune system, immunodeficiency can be caused by numerous and different causes, and it can involve acquired or innate immunity, both in the humoral and cellular components, as follows: innate pathologies (e.g., agammaglobulinemia linked to the X sex chromosome, one common variable immunodeficiency, severe combined immunodeficiency, DiGeorge syndrome, and congenital hypogammaglobulinemia), systemic diseases (e.g., autoimmune diseases, diabetes, chronic infections, and solid and liquid malignancies, such as leukemia, lymphoma, and multiple myeloma), and pharmacological therapies (e.g., chemotherapy, antirheumatics, immunosuppressants, and glucocorticoids), which are the main causes of immunodeficiency [8,9].

By definition, immunodeficiency is characterized by a functional deficit of the immune system (either congenital or acquired). Immunosuppression is a pathological condition characterized by the inhibition of one or more components of the immune system (natural or acquired), and it occurs following an intercurrent disease or autoimmune pathologies [10]. Immunosuppression also refers to pharmacological treatment with immunosuppressive drugs capable of inhibiting an immune system response [11]. Therefore, immunocompromised patients have a reduced ability to fight infections and other diseases.

Numerous studies have shown that in immunosuppressed subjects, there is a higher incidence of cancers than in a population with normal immunity [12]. The increased susceptibility to infections (i.e., HPV, candida, Helicobacter pylori, etc.) and the reduced immune response to infections in immunosuppressed subjects could represent a further mechanism that favors the onset of neoplasms. Furthermore, immunosuppression is, at the same time, one of the risk factors for the onset of oncological pathologies, but it is also a condition that could favor the loco-regional and distant growth and spread of cancers. In fact, the literature shows that immunosuppression is not only a risk factor for the genesis of a cancer but also a factor for the prognosis of its course [13].

Although the relationship between immunosuppression and the increased incidence of systemic cancers is now well documented, currently, it is not clear how much the risk of developing oral cancer increases in immunosuppressed subjects and what effect immunosuppression has on prognosis in terms of survival. The purpose of this systematic review was, therefore, to evaluate the association and the possible correlation between the state of depression of the immune system and the development of oral cancer through the evaluation of the incidence of oral cancer in patients with systemic immunosuppression and to compare that to data from official databases (Globocan, WHO), which lacked precise data on non-immunosuppressed subjects.

## 2. Materials and Methods

In the present systematic review, the adopted protocol followed the Preferred Reporting Items for Systematic Reviews and Meta-Analyses (PRISMA) statement. The review protocol was registered in PROSPERO database (CRD42021243898).

### 2.1. PICOS Question

The following question was developed according to the population, intervention, comparison, outcome, and study design (PICOS).

Population: immunosuppressed patients who later developed oral cancer were included in this systematic review.

Intervention: patients with systemic immunodepression due to various factors (immunodepression, malnutrition, infections, autoimmune diseases, genetic immunosuppression, immunosuppression as a consequence of immunosuppressive therapy or radiotherapy, and oncologic immunosuppression) who subsequently developed oral cancer were considered.

Comparison: the rates of development of oral cancer in non-immunosuppressed patients and the rates of development of oral cancer in immunosuppressed patients were compared.

Outcome: the primary outcome was to evaluate the incidence of oral carcinoma in immunosuppressed patients.

Study design: cohorts, case controls, cross-sectional studies, and randomized clinical trials (RCTs) with no fewer than 10 patients were included. All case reports, case series with less than 10 patients, in vitro or in vivo studies based on animals, systematic reviews, letters to the editor, cases of oral cancer related to human papilloma virus (HPV), and articles published in languages other than Italian, English, and Spanish were excluded.

### 2.2. Focused Question

The question on which attention was focused was formulated on the basis of the PICOS criteria: “Do immunosuppressed patients have a higher rate of development of oral cancer than healthy patients?”.

### 2.3. Research

The research was conducted on three databases (MEDLINE via OVID, Scopus, and Web of Science) from the start of their activity in May 2022, using a combination of key words and MeSH terms as follows: ((immunosuppression OR malnutrition OR infections OR autoimmune disease OR X-linked agammaglobulinemia OR common variable immunodeficiency OR selective immunoglobulin A deficiency OR hyper IgM syndrome OR DiGeorge syndrome OR severe combined immunodeficiency OR Wiskott–Aldrich syndrome OR acquired immunodeficiency syndrome OR AIDS OR immunosuppressive therapy OR radiotherapy OR “other systemic cancers” OR leukaemia OR lymphoma) AND “Oral Cancer”), (“Oral Carcinoma” AND (immunosuppression OR malnutrition OR infections OR autoimmune disease OR X-linked agammaglobulinemia OR common variable immunodeficiency OR selective immunoglobulin A deficiency OR hyper IgM syndrome OR DiGeorge syndrome OR severe combined immunodeficiency OR Wiskott–Aldrich syndrome OR acquired immunodeficiency syndrome OR AIDS OR immunosuppressive therapy OR radiotherapy OR “other systemic cancers” OR leukaemia OR lymphoma)), and (“Oral Neoplasms” AND (immunosuppression OR malnutrition OR infections OR autoimmune disease OR X-linked agammaglobulinemia OR common variable immunodeficiency OR selective immunoglobulin A deficiency OR hyper IgM syndrome OR DiGeorge syndrome OR severe combined immunodeficiency OR Wiskott–Aldrich syndrome OR acquired immunodeficiency syndrome OR AIDS OR immunosuppressive therapy OR radiotherapy OR “other systemic cancers” OR leukaemia OR lymphoma)). The date of the last search was 28 April 2023.

### 2.4. Manual Search

A manual search of articles published between 2002 and 2022 in the following peer-reviewed journals was performed: *Oral Oncology*, *Oral Diseases*, *Lancet Oncology,* and *Journal of Hematology and Oncology*.

### 2.5. Search of Unpublished Articles

Unpublished literature was searched in the U.S. National Institutes of Health clinical trials registry and the European Multidisciplinary Database to identify incumbent studies and grey literature. In addition, bibliographic references of all included articles and reviews were similarly checked to identify additional potentially relevant studies and increase the sensitivity of the search.

### 2.6. Study Selection

Based on the inclusion criteria, two authors independently and in duplicate (D.M. and F.S.) analyzed the titles and abstracts of the articles found. The authors retrieved the full versions of articles whose titles and abstracts appeared to meet the inclusion criteria or those, which reported insufficient data to make a clear decision. Next, the two authors independently read the full texts to determine whether the articles met these criteria. In cases where the two authors disagreed, agreement was sought through a comparison between the two, and when a solution could not be reached, a third senior author (R.P.) stepped in.

To calculate the agreement between the reviewers, Cohen’s kappa coefficient was used. The level of agreement was considered excellent when k was greater than 0.75, fair to good when it was between 0.40 and 0.74, and poor when it was less than 0.4 [14].

All articles that met the inclusion criteria were subjected to data extraction and quality assessments. All irrelevant articles were excluded, and the reasons for exclusion were as described.

### 2.7. Extraction Data

The data were collected using a purpose-built data extraction form. In cases where the publication did not provide all the necessary data, the corresponding author was contacted by e-mail to obtain the missing data. In the event that the two authors disagreed about one of the publications, a discussion was opened, which, in cases of disagreement, required the intervention of the third author.

In cases of redundant publications, the most recent article and the one with the largest follow-up were included.

### 2.8. Quality Assessment

The risk of bias in the included studies was independently assessed in duplicate by two authors as part of the data extraction process.

An assessment of risk of bias was undertaken using the Newcastle–Ottawa Scale (NOS) [15]. The presence of each parameter was recorded with a green mark, while absence was recorded with a red mark (0). Papers with 1–3 green marks were classified as high risk of bias, those with 4–6 green marks were classified as medium risk, and those with 7–9 green marks were classified as low risk. A supplemental analysis was performed independently by the two examiners regarding the overall quality of the evidence for any performed meta-analysis using the Grading of Recommendations, Assessment, Development, and Evaluations (GRADE) system [16]. Any disagreement between the two reviewers (D.M. and F.S.) was solved by discussion with the author supervisor (R.P.).

Publication bias was assessed through a funnel plot, which was made using Excel software (Microsoft Excel^®^).

### 2.9. Heterogeneity Assessment

The OpenMeta software was used for assessing the heterogeneity of the studies included in any conducted meta-analysis (OpenMeta, Inc.©, Zug, Zug, Switzerland). The authors calculated the comparability of the observed proportions across the results with chance alone using the I2 test. In cases where the *p*-value was <0.1, the heterogeneity was considered significant. Moreover, the same test was considered as a measure of heterogeneity across studies, following the subsequent scheme [17]: 0–40%, negligible; 30–60%, moderate; 50–90%, substantial; and 75–100%, considerable.

### 2.10. Data Analysis

Descriptive characteristics of the studies are expressed as means/medians and/or frequencies, as appropriate, depending on the variables.

Meta-analyses were performed only when there were studies comparing similar groups and reporting the same outcomes. In such cases, the meta-analyses were performed with a fixed-effect model. A random-effect model was used only in the case of not-negligible heterogeneity across the included studies (>50%).

A forest plot was created to illustrate the effects on the meta-analysis of individual studies and the overall estimate. OpenMeta-analyst [18] was used to perform all analyses. The cut-off value of significance was set at *p* < 0.05.

## 3. Results

### 3.1. Study Selection

A flowchart of the search strategy and study selection is shown in Figure 1.

A total of 2843 articles was identified, with 2796 found through electronic searches and 47 found through other sources. Out of the 2709 studies that resulted after removal of the duplicates, 2470 were excluded as a result of title and abstract reading (inter-reader agreement, k = 0.78). Eventually, out of the 239 articles that remained to be evaluated in the full-text, 44 met the inclusion criteria and were included in either the qualitative or quantitative analyses (meta-analysis); in contrast, 195 were excluded. All information about full-text articles excluded, with reasons are included in the Appendix A.

### 3.2. Study Characteristics

The characteristics of the included studies are summarized in Table 1.

Both prospective (five studies) and retrospective (nine studies) cohort studies were included in the review. Twenty-four studies presented data from national registries, and therefore, they were analyzed separately. In addition, six studies presented results related to a single immunosuppression condition, namely, graft-versus-host disease, and for this reason, they were analyzed separately, as this condition is, itself, a potentially malignant disorder of the oral cavity. All studies were conducted in an institutional environment.

### 3.3. Assessment of the Risk of Bias

The risk of bias is summarized in Figure 2 and Figure 3. The methodological quality of the included studies was high for 12 studies [19,22,24,25,28,31,32,33,36,51,53,58], moderate for 26 studies [20,21,23,25,26,29,30,32,34,37,38,39,40,41,42,43,44,45,46,47,48,49,50,52,54,55,56,57,60,61,62], and low for six studies [23,26,27,35,48,59].

The results regarding publication bias are presented in Figure 4, Figure 5 and Figure 6. Significant publication bias was found in the studies that presented results related to Graft Versus Host Disease (GVHD) and the national registries. The Grading of Recommendations, Assessment, Development, and Evaluations (GRADE) system provided information on the certainty of the conclusions and the strength of the evidence (Table 2). Although the meta-analyses drew conclusions from cohort studies, which are considered to be among the best-available evidence, they were considered to have only moderate strength of evidence because of the presence of at least one study with a high risk of bias and very wide confidence intervals.

### 3.4. Results of the Meta-Analyses

As reported earlier, three separate meta-analyses were conducted. The meta-analysis related to the national registries (Figure 7) was conducted on 23 studies with a total of 5,227,567 patients and found an “untransformed proportion” (PR) of 0.2% (95% CI: 0.002–0.003) (*p*-value of <0.001).

The meta-analysis concerning data not derived from the national registries (Figure 8) was conducted on 15 studies with a total of 6997 patients and found an “untransformed proportion” (PR) of 1% (95% CI: 0.006–0.014) (*p*-value of <0.001).

The meta-analysis regarding data about GVHD (Figure 9) was conducted on six studies with a total of 49,285 patients and found an “untransformed proportion” (PR) of 0.3% (95% CI: 0.001–0.005) (*p*-value of < 0.001).

The meta-analyses conducted on the three groups of patients revealed a general increased risk of developing an oral cancer in immunosuppressed populations. Such risk ranges from 0.2% to 1% depending on whether data from national registries are considered. In immunosuppressed patients, this evidence emphasizes the need to provide for a careful follow-up of suspicious lesions and potentially malignant disorders of the oral cavity.

## 4. Discussion

### 4.1. Summary of the Main Findings

The close relationship between the immune system and cancer immunoediting has been documented for many years for numerous cancers, including oral carcinoma, and this systematic review highlighted an incidence of oral carcinoma in immunosuppressed patients of 200 new cases per 100,000. If this is compared to data from registries on the incidence of oral cancer in the general population, which is approximately 4.1 per 100,000 subjects (ASR incidence = 4.1 per 100,000), immunosuppressed subjects have a risk of developing oral cancer that is 50 times higher than the general population. These raw data emphasize the need to establish clinical protocols for primary prevention and screening in all immunosuppressed subjects, likely with tailor-made protocols that depend on the cause of immunosuppression and the severity of the immunosuppression.

Some considerations of a methodological nature that emerged from this systematic review should be made in light of the literature. A first consideration concerns the definition that is used for patients with disorders of the immune system. The terms immunosuppression and immunodeficiency are often used interchangeably. This confusion is related to the subtle nuance that separates them. It could be specified that immunosuppression identifies a medical condition involving a general malfunction of the immune system, whereas immunodeficiency classifies the severity of this deficit into primary and secondary in relation to the cause. Furthermore, an aspect still unresolved concerns the identification of clinical and/or instrumental parameters that can identify the state of immunosuppression (considering both innate and acquired immunity, both cellular and humoral) and classify it in relation to the severity of the immunosuppression.

The present systematic review demonstrated that immunosuppression should be considered a risk factor for the development of oral cancer, with a percentage of increased risk ranging from 0.2% to 1% (95% CI: 0.2% to 1.4%). Considering the main causes of immunosuppression reported in the selected articles, there are some interesting considerations. In fact, in this systematic review, the authors decided to divide the results from the included papers into three main groups: the results derived from the literature analysis of the main reasons for immunosuppression (not from national registries), those from articles related to GVHD, and those from the registry analysis, which depict an increased risk of 1% (95% CI: 0.6% to 1.4%), of 0.3% (95% CI: 0.1% to 0.5%), and of 0.2% (95% CI: 0.2% to 0.3%), respectively. Articles referring to states of malnutrition were not included in this review, as they did not report adequate information regarding immune status.

### 4.2. Organ Transplantation

Organ transplantation, in particular, kidney transplantation, represents one of the main causes of immunosuppression most frequently associated with the onset of oral cavity neoplasms. The increased life expectancy of transplant recipients exposes them to prolonged immunosuppressive therapy (mainly cyclosporine), which is necessary to avoid the phenomenon of transplant rejection. In the study conducted by López-Pintor [62], 500 kidney transplant patients were recruited, and during follow-up, six cases of oral cancer were reported out of 500 patients (incidence of 1 patient per 100 subjects).

The same trend was seen for patients undergoing heart transplantation (HTx). Due to new techniques introduced in transplant surgery, survival after heart transplantation has improved significantly in recent decades. In the study conducted by Jääma-Holmberg (2019) [25], the risk of oral cancer after organ transplantation was two to four times higher than that of the general population, becoming one of the main long-term complications in this group of patients. Furthermore, it would appear that oral cancer occurs with a higher frequency in subjects who have undergone thoracic organ transplantation rather than those who have undergone abdominal organ transplants (i.e., liver and kidney). This different risk of oral cancer in relation to the type of organ transplanted could be partly related to the different pharmacological regimens adopted and partly linked to the underlying pathologies that lead to the need for transplants. Further studies should stratify the risk of oral cancer in relation to the type of organ transplanted.

### 4.3. Other Cancers

Another cause of immunosuppression associated with a greater risk of developing oral cavity cancer is represented by the treatment of thyroid neoplasms. The number of newly diagnosed cases of thyroid cancer has increased in recent years due to technological advances and the spread of cytological tests for early diagnosis. Patients who underwent partial or total thyroidectomy and those who received radio-iodine treatment for the treatment of thyroid cancer reported an increased risk of developing oral cancer. The study by Hsu et al. (2014) [40] showed an increased association between thyroid cancer and subsequent head and neck cancer. This association found that its biochemical-molecular explanation was related to the intrinsic carcinogenic action of radio-iodine, which can possibly be enhanced by pre-existing molecular genetic mutations in a framework of immunological impairment linked to the partial or total removal of the thyroid.

### 4.4. Infectious Agents

Other known causes of immunosuppressive states are infectious agents (i.e., HCV, HIV, and HPV). This literature review reported only one study, which was conducted by Su et al. [49] that highlighted an incidence of 698 cases of carcinoma out of 147,962 patients. The risk of oral cancer appears to be lower in HCV patients receiving pegylated interferon (PEG-IFN) therapy than that of untreated HCV patients. Further studies should investigate the role of HCV infection in oral cancer oncogenesis, with particular attention paid to the type of therapy administered to patients.

Studies investigating the role of HIV as a cause of immunosuppression were not included in this review. In fact, it is known that HIV infection causes a depletion of CD4+ T lymphocytes, with consequent impairment of the immune system. Acquired immunodeficiency could, therefore, lead to an increased risk of oral cancer. The study conducted by Precious K. Motlokwa et al. (2022) on an oral cancer population in sub-Saharan Africa did not show an increased risk of carcinogenicity in a group of HIV-infected patients [63]. This could be partly explained by new antiretroviral therapies, which allow clinicians to gain control of HIV infections and, therefore, reduce the impairment of patients’ immune systems. Further studies are needed to evaluate whether there is a real risk in HIV-positive patients and whether there are associated risk factors (CD4 T lymphocyte count or traditional antiretroviral therapies vs HAART).

### 4.5. Hematopoietic Stem Cell Transplantation (HSC)

Within the selected articles, it was possible to identify a group of articles conducted on patients undergoing hematopoietic stem cell transplantation (HSC), which now represents an essential therapy for the treatment of various haemato-lymphoproliferative diseases and other benign conditions (multiple myeloma, lymphomas, autoimmune disorders, etc.). In the study conducted by Santarone et al. (2020) [56], patients undergoing HSC transplantation reported the incidence of developing a malignancy at double the rate of the general population. In support of this, Dyer and colleagues [54] also found a similar incidence rate in patients undergoing HSC transplantation, underlining the importance of regular follow-ups with patients.

Furthermore, GVHD is among the adverse events associated with HSC transplantation. This clinical condition represents an adverse immunological phenomenon following HSC transplantation. GVHD oral lesions are among the so-called potentially malignant disorders, as they have a greater risk of neoplastic degeneration than healthy mucosa. Furthermore, the most frequently used therapy in the treatment of GVHD involves the use of immunosuppressive agents (e.g., both topical and high potency systemic corticosteroids and calcineurin inhibitors), which, although they reduce the inflammatory component of GVHD lesions, could increase the risk of developing a secondary malignancy. The risk of developing malignancy in patients with chronic GVHD was significantly increased compared with the general population, with a standard incidence ratio (SIR) of 1.8 and a 95% confidence interval (95% CI) of 1.5–2.0. The risk is much higher for cancer of the oral cavity (SIR = 15.7, 95% CI, 12.1–20.1), cancer of the esophagus (SIR = 8.5, 95% CI, 6.1–11.5), colon cancer (SIR = 1.9, 95% CI, 1.2–2.7), skin cancer (SIR = 7.2, 95% CI, 3.9–12.4), and cancers of the nervous system (SIR = 4.1, 95% CI, 1.2–8.4). The risk of developing oral, esophageal, or skin cancer appears to have a maximum incidence 1 year after transplantation [61].

### 4.6. Strengths and Limitations of the Present Systematic Review

Finally, the data obtained from this systematic review were partly extrapolated from the analysis of national registers from China, Japan, Republic of Korea, India, Taiwan, and Nordic Scandinavian countries. As these databases have a large amount of data, they can lead to significant statistical variations capable of creating very significant discrepancies in the results. In light of this, a meta-analysis dedicated solely to the analysis of the data obtained from these registries was conducted in this systematic review. It is also known that cancer of the oral cavity has a notably high incidence in the aforementioned countries (e.g., China and India) due to the different cultural and social habits. The funnel plot shown in Figure 4 revealed the presence of some studies with particularly discrepant data with respect to the confidence interval of the meta-analysis. Specifically, the study conducted by Levi et al. was discrepant to the funnel plot, and for this reason, it was removed from the statistical analysis and presented only in a qualitative form.

From a methodological point of view, all the studies included in this review had the main objective of investigating the incidence of cancer in other sites. Therefore, further prospective observational studies evaluating the occurrence of oral cancers in immunosuppressed patients as the main outcome while also taking into account the main risk factors of oral cancer that may influence this association (e.g., smoking, candida, HPV, and alcohol) are required. Moreover, it is essential to consider adequate follow-ups to avoid an underestimation of the real incidence of oral carcinomas. The time factor certainly plays an important role in the carcinogenic process, considering that a prolonged state of immunosuppression can increase the risk of the onset of neoplasms.

## 5. Conclusions

The results obtained from the systematic review indicated that immunosuppression is to be considered a risk factor for the development of oral cancer.

Particular attention and accurate follow-ups with all immunosuppressed patients are, therefore, essential in order to intercept clinical situations at an early stage that could evolve into oral cancer.

Further studies are needed to investigate the effective role of immunosuppression in carcinogenesis and to identify any risk factors.

## Figures and Tables

**Figure 1 cancers-15-03077-f001:**
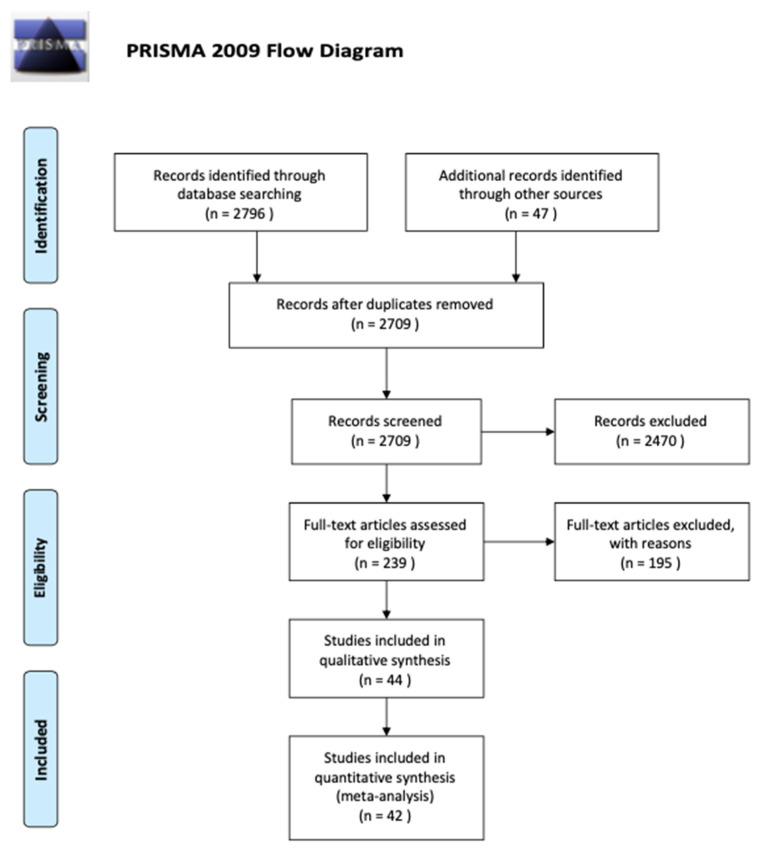
Flowchart of the selection of the studies for the review.

**Figure 2 cancers-15-03077-f002:**
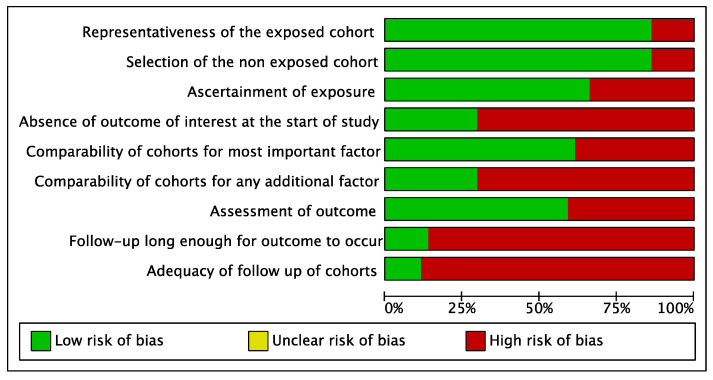
Risk of bias graph.

**Figure 3 cancers-15-03077-f003:**
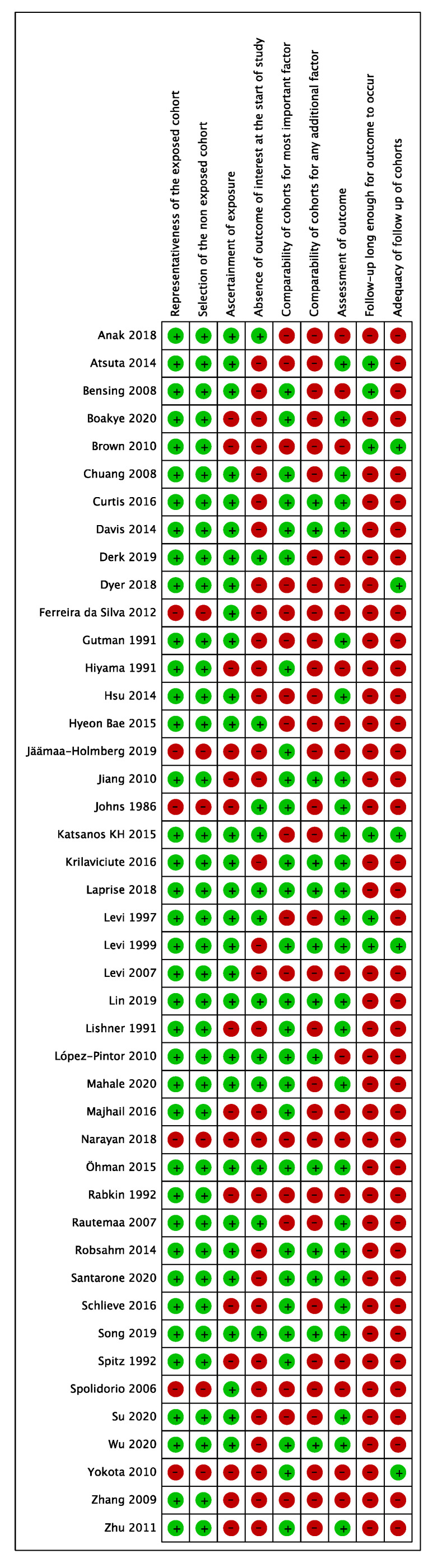
Risk of bias summary [19,20,21,22,23,24,25,26,27,28,29,30,31,32,33,34,35,36,37,38,39,40,41,42,43,44,45,46,47,48,49,50,51,52,53,54,55,56,57,58,59,60,61,62].

**Figure 4 cancers-15-03077-f004:**
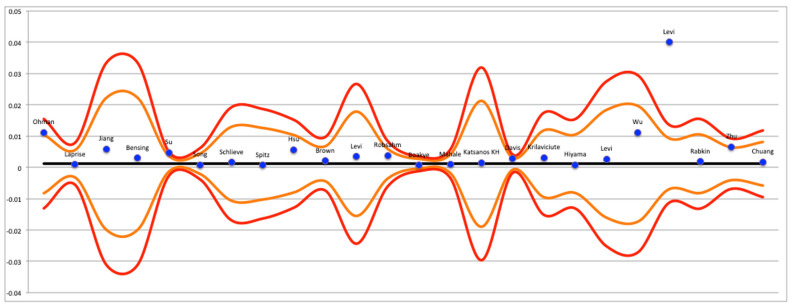
Funnel plot of studies with data from national registries [20,23,26,31,32,33,34,35,36,37,38,39,40,41,42,44,45,46,47,48,49,50,57,59].

**Figure 5 cancers-15-03077-f005:**
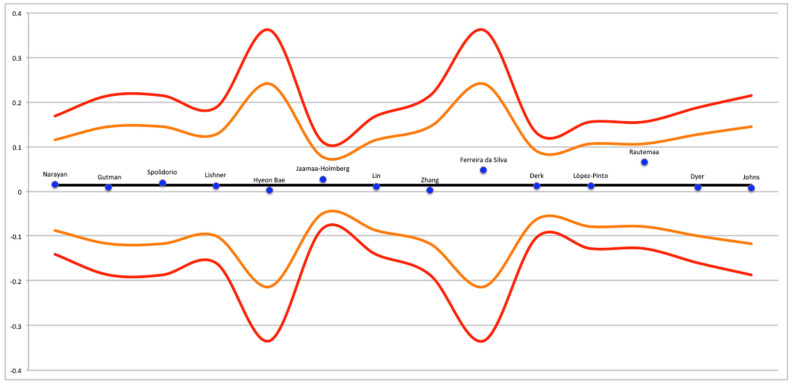
Funnel plot of studies with data not from national registries [19,21,22,24,25,27,28,29,30,43,54,58,60].

**Figure 6 cancers-15-03077-f006:**
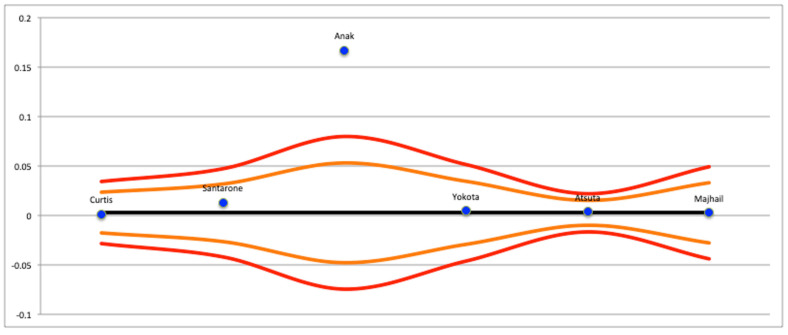
Funnel plot of studies with data about GVHD [51,52,53,55,56,61].

**Figure 7 cancers-15-03077-f007:**
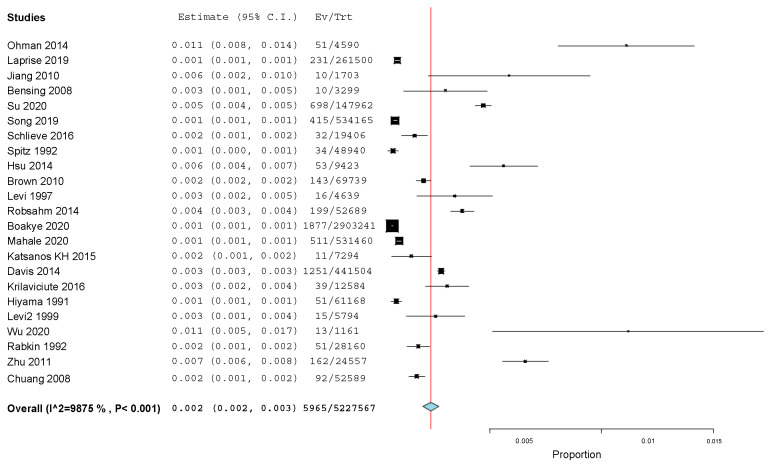
Meta-analysis related to data coming from national registries [20,23,26,31,32,33,34,35,37,38,39,40,41,42,44,45,46,47,48,49,50,57,59].

**Figure 8 cancers-15-03077-f008:**
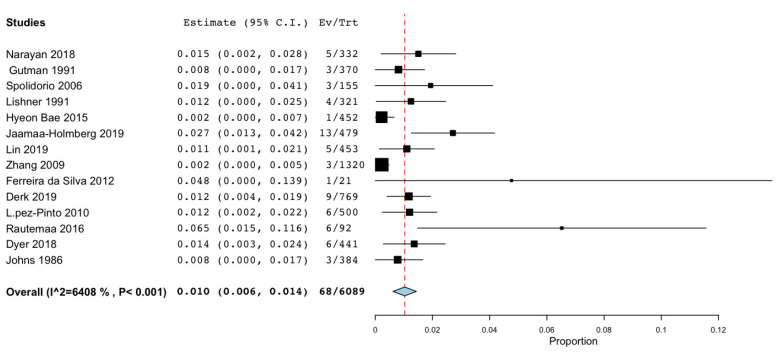
Meta-analysis related to data not coming from national registries [19,21,22,24,25,27,28,29,30,43,54,58,60,62].

**Figure 9 cancers-15-03077-f009:**
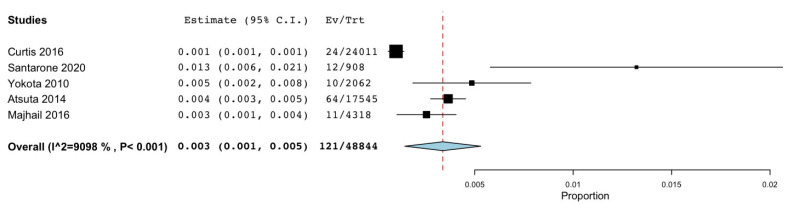
Meta-analysis regarding data about GVHD [51,52,53,56,61].

**Table 1 cancers-15-03077-t001:** Characteristics of the included studies.

Authors-Year	Study Setting	Study Design	No. Patients (Gender)	Cause of Immunodepression	No. Patients Who Delevoped Oral Cancer	% of Oral Cancer (Cancer/Tot)	Age (Mean)	Gender	Aim	Oral Cancer Site	Follow Up (Years)
				Organ Transplant							
Spolidorio, 2006 [19]	São Paulo Hospital	P	155 (120 M, 35 F)	Cyclosporin A or tacrolimus	3	1.93%	Unknown	NR	To determine the oral status of renal transplant recipients receiving cyclosporin A or tacrolimus as immunosuppressant	Lip	unknown
Jiang, 2010 [20]	Canadian Organ Replacement Register	R	1703 (1405 M, 298 F)	Heart transplantation	10	0.58%	54.4	NR	To assess the long-term risk of developing cancer among heart transplant recipients compared to the Canadian general population	NR	6.08 years
Lòpez-Pintor, 2010 [21]	Hospital Universitario 12 de Octubre, Madrid, Spain	R	500 (193 F, 307 M)	Renal transplantation	6	1.2%	57.33	M	To establish the incidence of lip cancer (LC) in a population of renal transplant patients (RTPs)	lip	18
Ferreira da Silva, 2012 [22]	Department of the federal university of Sergipe, Brazil	R	21 (7 F, 14 M)	Kidney transplantation	1	4.76%	42	M	To investigate oral lesions in kidney transplant patients	lip	2.5 (mean)
Ohman, 2014 [23]	Sahlgrenska University Hospital Register	R	4590 (2839 M, 1751 F)	Transplantation	51	1.11%	62	NR	To verify an increased risk of oral and lip cancer in solid organ transplantation patients	4 tongue, 5 salivary glands, 3 floor of mouth, 3 gingiva, palate, bucca, 34 lip	Median 6.3 years
Narayan, 2018 [24]	Medwin Hospitals, Telangana, India	P	332	Renal transplantation	5	1.50%	NR	NR	To identify the number of patients with renal transplant who developed second cancer	tongue	26
Jaamaa-Holmberg, 2019 [25]	NA	R	479 (381 M, 98 F)	Heart transplantation	13	2.71%	Unknown	NR	To demonstrate that cancer incidence in Finnish HTx-recipients is six times higher than in general Finnish population	7 lip, 4 tongue, 1 salivary glands, 1 non specified	Median 7.8 years
Laprise, 2019 [26]	The scientific Registry of transplant recipients	R	261,500 (174,475 M, 109,357 F)	Transplantation	231	0.09%	50	NR	To evaluate the incidence of lip cancer after solid organ transplantation	231 lip	Median 3.96 years
Lin, 2019 [27]	Changhua Christian Hospital	R	455-2 (453)	Liver transplantation	5	1.10%	56	1 F, 4 M	To identify the number of head and neck cancer in liver transplant recipients	3 tongue, 1 retromolar trigone, 1 buccal mucosa, 1 parotid gland	NR
				**Other** **Cancers**							
Johns, 1986 [28]	Johns Hopkins Medical Istitutions, Baltimore	R	384 (206 F, 178 M)	Salivary gland or thyroid gland malignancies	3	0.78%	NR	1 F, 2 M	To determine the exact risk of multiple primary neoplasms in patients with salivary gland or thyroid gland malignancies	3 salivary glands	10
Gutman, 1991 [29]	Tel Aviv Medical Center	P	370 (133 M, 237 F)	Melanoma	3	0.81%	60.5	F	To identify the number of patients with GVHD who developed second cancer	NR	Different based on stages
Lishner, 1991 [30]	Princess Margaret Hospital, Toronto	R	321	Non-Hodgkin’s lymphoma	4	1.24%	48	3 M, 1 unknown	To evaluate the incidence of second malignant tumors in patients with Non-Hodgkin’s lymphoma	3 tongue, 1 gingiva	At least 6 months
Hiyama, 1991 [31]	Department of field research, Osaka	R	61,168 (22,391 F, 38,777 M)	Stomach cancer	51	0.08%	NR	NR	To determine the risk of second primary cancer after diagnosis of stomach cancer in Osaka	NR	30
Spitz, 1992 [32]	National Cancer Institute	R	48,940 (F)	Cervix cancer	34	0.07%	NR	F	To evaluate the association between malignancies of the upper aerodigestive tract and uterine cervix	NR	11 years
Rabkin, 1992 [33]	National cancer institute, Belthesda	R	28,160 (25,295 F, 2865 M)	Anal and cervical carcinoma	51	0.18%	NR	NR	To determine the risk of second primary cancer following anal and cervical carcinoma	NR	NR
Levi, 1997 [34]	The Cancer Registries, Switzerland	R	4639	Skin Cancer	16	0.34%	74	NR	To evaluate the incidence of second primary cancers in patients with skin cancer	5 lip, 3 salivary gland, 8 mouth	23 years
Levi, 1999 [35]	University of Milan, Italy	R	5794	Lung carcinoma	15	0.26%	NR	NR	To determine the risk of second primary cancer in patients with lung carcinoma	NR	22
Levi, 2007 [36]	Universitè de Lausanne	R	1672 (424 F, 1248 M)	Esophageal cancer	67	4.00%	55	NR	To determine the risk of second neoplasms after esophageal cancer	NR	30
Chuang, 2008 [37]	Lyon, France	R	52,589 (19,110 F, 33,479 M)	Esophageal cancer	92	0.18%	NR	NR	To assess the risk of second primary cancers following a first primary esophageal cancer	NR	10
Brown, 2010 [38]	The National Cancer Institute’s Survival	R	69,739 (F)	Endometrial cancer	143	0.20%	62	F	To examine the risk of subsequent primary malignancies (SPMs) in women diagnosed with endometrial cancer.	NR	11.2 years
Zhu, 2011 [39]	Academy of Medical Sciences, Gansu, China	R	24,557 (6253 F, 18,304 M)	Treatment of esophageal cancer	162	0.66%	NR	NR	To determine the risk of second primary cancer after treatment for esophageal cancer	NR	34
Hsu, 2014 [40]	Taiwan’s National Health Insurance	R	9423 (1940 M, 7483 F)	Thyroid cancer	53	0.56%	NR	NR	To determine the association of thyroid cancer with other malignancies in Taiwan.	40 mouth, 13 salivary glands	NR
Robsahm, 2014 [41]	Cancer Registry of Norway	R	52,689 (28,069 CMM, 24,620 SCC)	Squamous cell carcinoma and melanomas	47 (CMM), 152 (SCC)	0.37%	NR	33 M, 14 F (CMM)/114 M, 38 F (SCC)	To examine the risk of a new primary cancer following an initial skin cancer	NR	10.1
Davis, 2014 [42]	University of Michigan Medical school	R	441,504 (M)	Prostate cancer	1251	0.28%	NR	NR	To determine the risk of second primary tumors in men with prostate cancer	NR	10
Hyeon Bae, 2015 [43]	Chonnam National University Hospital, Hwasun, Korea	R	452 (208 M, 244 F)	Melanoma	1	0.22%	Unknown	NR	To assess the presence of other primary cancer in patients with acral and non-acral melanomas	NR	No
Krilaviciute, 2016 [44]	National cancer institute, Vilnius, Lithuania	R	12,584 (8074 F, 4510 M)	Basal cell carcinoma	39	0.31%	NR	NR	To determine the risk of second primary cancer in basal cell carcinoma patients in Lithuania	14 lip, 25 other in oral cavity	14
Schlieve, 2016 [45]	University of Tennessee	R	19,406/849	Primary Non-head-neck cancer	32	80%/	67	NR	To determine the rate of second primary head and neck cancer development among patients with a primary cancer diagnosed outside of the head and neck region, to present the clinical characteristics of this population, and to determine if any variables are associated with survival.	11 gingiva, 7 tongue, 4 base of tongue, 4 buccal, 3 floor of mouth, 2 palate, 1 parotid	10 years
Boakye, 2020 [46]	National Cancer Institute’s Surveillance	R	2,903,241	First primary cancers	1877	0.064	63.1	1303 M, 574 F	To describe the risk of developing a second primary cancer among survivors of 10 cancer sites with the highest survival rates in the United States	1462 tongue, 343 floor, 72 salivary glands	3.8 years
Wu, 2020 [47]	People’s hospital of Nanjing, China	R	1161 (542 F, 619 M)	Pulmonary high-grade neuroendocrine carcinoma	13	1.12%	NR	NR	To determine the risk of second primary cancer in patients with pulmonary high-grade neuroendocrine carcinoma	floor of mouth, and gum and other mouth	16
				**Infectious Diseases**							
Song, 2019 [48]	The China Kadoorie Biobank	R	(a) 496,732 (203,660 M, 294,072 F) (b)37,336 (c) 97 (73 M, 24 F)	HBV	(a) 415 (b) no cases c) NR	(a) 1.98%/0.08% (b) no c) NR	(a) 51.5 (b)	NR	To assess the association between chronic HBV infection and risk of all cancer types	NR	(a) 8.85 (b)
Su, 2020 [49]	National Health insurance Research Database	P	100,058 (50,029 HCV-50,029 NO HCV) + 47,904 (23 952 therapy-23,952 no therapy)	HCV and anti-HCV therapy	229 (NO-HCV) 265 (HCV) + 146 (no therapy) 58 (therapy)	0.47%	59 (1 group)- 51 (2 group)	NR	To investigate the association between chronic hepatitis C and oral cancer, and the development of oral cancer after anti-hepatitis C virus (HCV) therapy	NR	7.9 years non-HCV/5.1 years HCV + 4.9 years no therapy/3.4 years therapy
Mahale, 2020 [50]	Surveillance, Epidemiology, and End Results (SEER)	R	531,460 (384,777 M, 146,683 F)	HIV+/lymphoid malignancies	511	0.01%	NR	NR	To describe the risk of cancers following lymphoid malignancies among HIV-infected people.	NR	NR
				**HSC**							
Yokota, 2010 [51]	Kanto Study Group for Cell Therapy	R	2062 (1225 M, 837 F)	Allogeneic hematopoietic SCT	10	35.7%/0.48%	42	5 M, 4 F, 1 Unknown	To evaluate the incidence and risk factors for secondary solid tumors in Japan after hematopoietic SCT	5 tongue, 3 gingiva, 2 oral mucosa	Median 5.7 years
Curtis, 2016 [52]	Center for International Blood and Marrow Transplant Research	P	24011	GVHD	24	13.11%/0.1%	NR	NR	To identify the number of patients with GVHD who developed second cancer	NR	30
Majhail, 2016 [53]	Center for International Blood and Marrow Transplant Research	R	4318 (2415 M, 1903 F)	Hematopoietic cell transplant	11	16.6%/0.25%	44	NR	To evaluate the risk of secondary solid cancers among allogeneic hematopoietic cell transplant recipients	NR	NR
Dyer, 2018 [54]	Blood and Marrow Transplant Network, Australia.	P	441 (191 F, 250 M)	Blood and marrow transplant	4	1.5%	NR	NR	To investigate oral health in blood and marrow transplant recipients	NR	12
Anak, 2018 [55]	Istanbul University Faculty of Istanbul Medicine, Our Children Leukemia Foundation BMT Center	P	24 (12 M, 12 F)	Hematopoietic cell transplantation in Fanconi Anemia patients	4	21	NR	To investigate SCC development after HSCT and examine features of the follow-up patients	4 retromolar trigone	NR	NR
Santarone, 2020 [56]	Bone marrow transplant center, Ospedale civile, Pescara, Italy	R	908 (498 M, 410 F)	Hematopoietic cell transplantation	12	100%/1.32%	47	8 M, 4 F	To demonstrate that oral cGVHD and a diagnosis of non-malignant hematologic disease are strong risk factors in the SOC development	6 tongue, 1 lower lip, 3 cheek mucosa, 1 gingival fornix, 1 hard palate	Unknown
				**Inflammatory** **Diseases**							
Bensing, 2008 [57]	National Death Register/Swedish Cancer Register	R	3299 (1359 M, 1940 F)	Autoimmune primary adrenocortical insufficiency	10	0.30%		NR	To assess the increased death risk and altered cancer incidence in patients with autoimmune primary adrenocortical insufficiency	NR	29 years
Zhang, 2009 [58]	Peking Union Medical College Hospital	R	1320 (1201 F, 119 M)	Sjögren’s syndrome	3	10%	50.7	NR	To identify the incidence of malignancy in primary Sjögren’s syndrome	2 tongue, 1 parotid gland	4.4 (mean)
Katsanos KH, 2015 [59]	Clinical Gastroenterology and Hepatology, NT.; USA	R	7294 (3785 F, 3509 M)	Inflammatory bowel disease (IBD)	11	0.15%	44.6	4 F, 7 M	To identify the number of patients with IBD that developed oral cancer	6 tongue, 2 hard palate, 3 buccal	NR
Rautemaa, 2016 [60]	Helsinki Hospital, Finland	R	92 (47 F, 45 M)	APECED	6	6.52%	37	2 F, 4 M	To study the possible association of APECED with oral and esophageal carcinoma.	buccal mucosa	NR
Derk, 2019 [21]	Thomas Jefferson University Philadelphia, Pennsylvania, USA	P	769	Systemic sclerosis	9	1.17%	49.2	NR	To describe the incidence of carcinoma of the tongue in a cohort of patients with systemic sclerosis	tongue	16
				**NR**							
Atsuta, 2014 [61]	Transplant Registry Unified Management Program	R	17545 (10,386 M, 7149 F)	NR	64	23.80%	NR	NR	To determine the incidence and the risk factors for secondary solid tumors after allogenic stem cell transplantation	NR	NR

NR: Not reported, P: Prospective, R: Retrospective.

**Table 2 cancers-15-03077-t002:** GRADE summary of findings for meta-analysis on immunosuppression and oral cancer incidence.

Quality Assessment, Outcome: Oral Cancer Incidence in Patients with Immunosuppression						
Question: Does the Immunosuppression Condition Have Influence on Oral Cancer Incidence?						
Number of Studies according to meta-analysis	Study design	Risk of Bias	Inconsistency	Indirectness	Imprecision	Publication bias
Meta-analysis on data from national registers (Figure 7): 23 studies	Cohort studies	Serious	Serious ^a^	Not Serious	Serious ^b^	Detected (1 study)
Meta-analysis on data not from national registers (Figure 8): 14 studies	Cohort studies	Serious	Not Serious	Not Serious	Serious ^b^	Undetected
Meta-analysis on GVHD patients (Figure 9): 5 studies	Cohort studies	Serious	Serious ^a^	Not Serious	Serious ^b^	Detected (1 study)

^a^. Due to high heterogeneity across studies. ^b^. Due to wide confidence intervals.

## Data Availability

Data are available upon request to the corresponding author.

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
