# Peer review of "Is Systemic Immunosuppression a Risk Factor for Oral Cancer? A Systematic Review and Meta-Analysis"

_cancers, 2023, doi:10.3390/cancers15123077_

Round 1
Reviewer 1 Report
1. In Table 1, I would recommend grouping studies by reason of immunosuppression: transplantation, cancer, etc.
2. There are very small signatures on the drawings, almost unreadable. Gotta fix it.
Author Response
REVIEWER 1
1. In Table 1, I would recommend grouping studies by reason of immunosuppression: transplantation, cancer, etc.
Thanks for the comment. Table 1 presentation has been changed as suggested.
2. There are very small signatures on the drawings, almost unreadable. Gotta fix it.
Thanks for the comment. It was a little tricky to understand which drawings the reviewer was referring to but at last we identified figures from 3 to 6 probably to be the ones. Such figures have been enlarged hoping this would not contravene editorial policies.
Reviewer 2 Report
Review comments for “Is systemic immunosuppression a risk factor for oral cancer? A systematic review and meta-analysis.”
Major comments:
1. First paragraph of Introduction: It’s great that you introduced the number of annual cases of oral cancer and the risk factors of oral cancer. However, it would be helpful to include an introduction to the current clinical treatment for oral cancer. This will provide readers with a more comprehensive understanding of oral cancer.
2. Line 65: Expand on immunosurveillance and cancer immunoediting: Since these concepts are mentioned briefly, you could provide a bit more explanation or context to help readers understand their significance and relevance to the topic.
3. Introduction: The term "cancer immunoediting" is used to describe the evolution of tumors, wherein tumor cells become less effectively recognized and killed by the immune system. On the other hand, "immunosuppression" is a pathological condition characterized by the inhibition of one or more components of the immune system, whether natural or acquired. However, currently, there is no description illustrating the relationship between immunoediting and immunosuppression. Adding such an explanation would be beneficial.
4. 3.3. Results of the meta-analyses: Only the results were shown in this paragraph. It would be beneficial to add a conclusion to the paragraph, summarizing the key findings and providing insight into the purpose of the meta-analyses. This will help readers understand the significance of the results obtained.
5. Discussion: The discussion seems to be a compilation of findings from different studies, but it lacks a clear structure. It would be beneficial to organize the content into subsections based on the main themes or factors discussed, such as main causes of immunosuppression, associated risks, and implications for prevention and screening. This would make the discussion easier to follow and comprehend.
Minor comments:
1. Line 290: There are some abbreviations whose full name didn’t showed in the manuscript. It would be beneficial to add the full name for each abbreviation like GVHD or GRADE.
2. References: The font of reference 4, 24, 27 are different with others. It would be beneficial to modify it.
Author Response
Major comments:
- 1.First paragraph of Introduction:It’s great that you introduced the number of annual cases of oral cancer and the risk factors of oral cancer. However, it would be helpful to include an introduction to the current clinical treatment for oral cancer. This will provide readers with a more comprehensive understanding of oral cancer.
- Line 65:Expand on immunosurveillance and cancer immunoediting: Since these concepts are mentioned briefly, you could provide a bit more explanation or context to help readers understand their significance and relevance to the topic.
- Introduction:The term "cancer immunoediting" is used to describe the evolution of tumors, wherein tumor cells become less effectively recognized and killed by the immune system. On the other hand, "immunosuppression" is a pathological condition characterized by the inhibition of one or more components of the immune system, whether natural or acquired. However, currently, there is no description illustrating the relationship between immunoediting and immunosuppression. Adding such an explanation would be beneficial.
Thanks for all the comments. All the suggested paragraphs have been added in introduction and the relative sections have been highlighted in yellow.
- 3.3. Results of the meta-analyses:Only the results were shown in this paragraph. It would be beneficial to add a conclusion to the paragraph, summarizing the key findings and providing insight into the purpose of the meta-analyses. This will help readers understand the significance of the results obtained.
Thanks for you comment. All the suggested information has been added in a paragraph at the end of the result section.
- 5.Discussion: The discussion seems to be a compilation of findings from different studies, but it lacks a clear structure. It would be beneficial to organize the content into subsections based on the main themes or factors discussed, such as main causes of immunosuppression, associated risks, and implications for prevention and screening. This would make the discussion easier to follow and comprehend.
Thanks for the comment. The discussion section has been divided in several subsections in order to increase the readers’ readability. All subsections’ titles have been highlighted in yellow.
Minor comments:
- Line 290:There are some abbreviations whose full name didn’t showed in the manuscript. It would be beneficial to add the full name for each abbreviation like GVHD or GRADE.
- References:The font of reference 4, 24, 27 are different with others. It would be beneficial to modify it.
Thanks for you comments. Abbreviations have been showed “in extenso” when reported for the first time and the font of reference number 4, 24 and 27 has been modified. Please note that, after the inclusion of some new paragraphs in introduction, the reference order could have changed.
Reviewer 3 Report
The topic of the present systematic review with meta-analysis, evaluating if immunosuppression should be considered a risk factor for oral cancer might be clinically relevant.
The review is well-conducted and the Methods are adequately described. However, I would suggest registering it on the PROSPERO register.
The submitted manuscript is well-organized, although some suggestion has been provided to the Authors.
Editing for the English language is not required.
The reviewer’s concerns are detailed in the pdf file attached.

Author Response
Dear reviewer, thanks for your comments. All the suggested references have been added. Inclusion criteria have been reported and a paragraph about the association of HPV and immunodeficiencies in oral carcinogenesis has been added. Moreover the paragraph about explanation of similarities and differences among immunosuppression and immunodeficiency has been shortened in introduction and enlarged in discussion section as suggested.
Round 2
Reviewer 3 Report
I congratulate the Authors for the work done